# Tetrandrine Treatment May Improve Clinical Outcome in Patients with COVID-19

**DOI:** 10.3390/medicina58091194

**Published:** 2022-09-01

**Authors:** Shiyin Chen, Yiran Liu, Juan Ge, Jianzhong Yin, Ting Shi, James Ntambara, Zhounan Cheng, Minjie Chu, Hongyan Gu

**Affiliations:** 1School of Medicine, Nantong University, Nantong 226000, China; 2Department of Epidemiology, School of Public Health, Nantong University, Nantong 226000, China; 3Institute of Geriatrics (Shanghai University), Affiliated Nantong Hospital of Shanghai University (The Sixth People’s Hospital of Nantong), Nantong 226000, China; 4Department of Respiratory, Daye Hospital of Chinese Medicine, Daye 435100, China

**Keywords:** COVID-19, SARS-CoV-2, tetrandrine, clinical outcome, traditional Chinese medicine

## Abstract

*Background and objectives:* The COVID-19 pandemic continues worldwide, and there is no effective treatment to treat it. Chinese medicine is considered the recommended treatment for COVID-19 in China. This study aimed to examine the effectiveness of tetrandrine in treating COVID-19, which is originally derived from Chinese medicine. *Materials and Methods:* A total of 60 patients, categorized into three types (mild, moderate, severe), from Daye Hospital of Chinese Medicine with a diagnosis of COVID-19 were included in this study. Demographics, medical history, treatment, and results were collected. We defined two main groups according to the clinical outcome between improvement and recovery. All underlying factors including clinical outcomes were assessed in the total number of COVID-19 patients and moderate-type patients. *Results:* In a total of 60 patients, there were significant differences in the clinical outcome underlying treatment with antibiotics, tetrandrine, and arbidol (*p* < 0.05). When the comparison was limited to the moderate type, treatment with tetrandrine further increased recovery rate (*p* = 0.007). However, the difference disappeared, and no association was indicated between the clinical outcome and the treatment with and without antibiotic (*p* = 0.224) and arbidol (*p* = 0.318) in the moderate-type patients. In all-type and moderate-type patients, tetrandrine improved the rate of improvement in cough and fatigue on day 7 (*p* < 0.05). *Conclusions:* Tetrandrine may improve clinical outcome in COVID-19 patientsand could be a promising potential natural antiviral agent for the prevention and treatment of COVID-19.

## 1. Introduction

The Corona Virus Disease 2019 (COVID-19) pandemic caused by severe acute respiratory syndrome coronavirus 2 (SARS-CoV-2) persists worldwide. By 1 August 2022, the total number of confirmed cases has reached nearly 600 million, and more than 6 million people have died from COVID-19. The pandemic situation in China was perfectly controlled by active control and intervention measures, with only a few areas suffering from localized epidemics. However, it looks like the global COVID-19 pandemic has been around for a long time, and the risk of transmission and spread will also remain. Meanwhile, SARS-CoV-2 keeps mutating during its pandemic, generating multiple variants. The Delta variant, prevalent in 2021, showed a stronger infection capacity and appears to be around 60% more transmissible than the highly infectious Alpha variant identified in the UK at the end of 2020 [1]. Compared to the Alpha variant, the Delta virus variant pathogenicity is greater, and COVID-19 patients infected with the Delta variant are at greater risk of hospitalization or emergency care [2,3]. Even worse, the dominate variant in 2022, Omicron, was found to have a higher transmission rate than all previous variants [4]. Therefore, there is an urgent need for an effective and specific antiviral treatment.

The most common symptoms of COVID-19 were non-specific and included mainly fever, cough, and muscle pain [5]. Lung damage consists of diffuse alveolar damage, exudative alveolar inflammation, lung cornification, interstitial pneumonia, pulmonary fibrosis, and focal bleeding [6]. COVID-19 can affect other parts of the body as well, such as the kidneys, liver, digestive system, cardiovascular system, and nervous system [7]. To date, clinical control of COVID-19 has been based only on symptom treatment from available therapeutic drugs and supportive treatments, including oxygen and mechanical ventilation.

There is still no effective therapy for COVID-19 and no clinically approved antiviral drugs for COVID-19. The therapeutic approaches now examined include antiviral and anti-inflammatory cytokines, anti-infectious and life-sustaining therapies, monoclonal antibodies, and passive immunotherapy [8]. Reusing existing compounds is a key point in the battle to develop potentially useful therapeutic approaches in record time [9].

Chinese herbal medicine, as a unique treatment in China, has a wide range of uses and healing properties for various diseases. Chinese medicine is considered the recommended treatment for COVID-19 in China. At present, many COVID-19 patients have been treated with Chinese medicine, and the overall healing effect is remarkable. Using Huoxiang Zhengqi dripping pills and Lianhua Qingwen granules in combination with Western medicine can improve the prognosis of patients with COVID-19 [10]. A systematic review that included seven randomized controlled trials (RCTs) showed that oral Chinese herbal medicine in combination with conventional Western therapy conferred better healing effects than conventional Western therapy, including Lianhua Qingwen capsules/granules, Jinhua Qinggan granules, Huoxiang Zhengqi dripping tablets, Toujie Quwen granules, and Lianhua Qingke granules [11]. These traditional Chinese medicines work mainly by enhancing immunomodulatory effects and relieving non-pulmonary clinical symptoms such as fever, cough, and fatigue [11,12]. Hence, certain traditional Chinese medicines may provide desperately sought relief from COVID-19.

It is worth mentioning that pulmonary fibrosis is involved almost all fatal COVID-19 cases. Zhan Xi et al. found that most patients with COVID-19 had varying degrees of post-inflammatory pulmonary fibrosis after discharge. The incidence of post-inflammatory pulmonary fibrosis was as high as 70% in patients with ordinary COVID-19 and even 100% in patients with severe pneumonia [13]. Most noteworthy, pulmonary fibrosis is also the main pathological changes of silicosis, which has been treated with tetrandrine for decades [14]. Tetrandrine is a traditional Chinese medicine originally obtained from *Stephania tetrandra* S. Moore, a plant of the genus Stephania in Menispermaceae, and used to treat rheumatism and arthralgia. The molecular formula of tetrandrine is C_38_H_42_N_2_O_6_, and its chemical structure is shown in Figure 1. Meanwhile, tetrandrine has clear anti pulmonary fibrosis effects, the main mechanisms of which may include antagonism of the calcium and calmodulin systems, inhibition of colonization, and differentiation of TGF-β1 fibrotic cells [15]. The combination of tetrandrine and acetylcysteine effervescent tablets can improve exercise tolerance, pulmonary function, clinical symptoms, and the chest X-ray findings of silicosis patients [16,17]. Thus, tetrandrine may specifically alleviate the pulmonary symptoms of COVID-19 via its anti-pulmonary fibrosis effects.

Previous research found that tetrandrine can block Ebola through its ability to block both two pore channel 1 (TPC1) and TPC2, as the endosomal calcium channels called TPCs appear to hold the responsibility for controlling movement of endosomes containing Ebola virus particles [18]. The ongoing studies of COVID-19 have discovered that endolysosomal two-pore cation channels have now emerged as potential novel targets for SARS-CoV treatment [19]. TPCs are intracellular calcium/cation channels located in the membranes of host endolysosomal compartments, which SARS-CoV-2, the virus causing COVID-19 (and several other viruses), depends upon for egress from these organelles and replication. Based on these data, many studies now suspect that tetrandrine can block the TPC2 in host cells and thus inhibit virus replication at low micromolar concentrations [20]. Further, tetrandrine was observed to have notable levels of synergy with remdesivir, which has demonstrated the most promising anti-viral therapeutic results [21]. Therefore, it is biologically plausible that tetrandrine may play a positive role in improving the clinical outcome of COVID-19.

Recently, Ou et al. constructed a pseudo-type lentiviral system using the optimized SARS-CoV-2 protein S and found that tetrandrine can prevent the entry of that viral via blocking TPC2 activity [22]. A previous study found that tetrandrine was a potential natural antiviral agent for the prevention and treatment of infection with HCoV-OC43, which is closely related to SARS-CoV and which shares several functional properties [23]. Although previous studies have focused on the mechanism level and indicated that tetrandrine could be a potential therapeutic agent against COVID-19 [24], there are no population-based studies on the actual effectiveness of tetrandrine in COVID-19 patients. Therefore, it is necessary to investigate the result of tetrandrine for clinical application in COVID-19. Therefore, based on the influence of different treatment schemes for COVID-19 patients in Daye Hospital of Chinese Medicine, this study intended to explore the actual clinical manifestations of tetrandrine in COVID-19 and provide epidemiological evidence for potential drug treatment schemes in COVID-19.

## 2. Materials and Methods

### 2.1. Study Population

The studies involving human participants were reviewed and approved by the Ethical Committee of the Sixth People’s Hospital of Nantong. The patients or their legal guardian provided written informed consent to participate in this study. This study enrolled 60 patients who were diagnosed with COVID-19 in Daye Hospital of Chinese Medicine in March 2020. The patients were classified into three types: mild, moderate, and severe, according to the National Health Commission of China’s Guidelines for Diagnosing and Treating COVID-19. The following data were taken from the hospital’s medical records: demographic information (age, gender), smoking status, BMI, initial severity of the disease, history of chronic disease (hypertension, diabetes, hepatitis, coronary artery disease), history of lung disease (tracheitis, bronchitis, asthma, bronchiectasis, obsolete pulmonary tuberculosis), treatment (antibiotic, hormone, phlegm reduction, supportive treatment, oxygen therapy, tetrandrine, ribavirin, vitamin C, interferon, Lianhua Qingwen, arbidol), length of hospital stay, time to remission, and result. All patients were categorized into two groups according to the outcome between improvement and recovery. The groups were compared in terms of age, gender, BMI, smoking status, clinical types, past medical history, treatment, and duration of hospitalization. Days from symptom onset to remission were recorded, and the association between treatment and clinical symptoms was compared by the day 7 improvement rate (Figure 2).

### 2.2. Chemicals

Tetrandrine tablets (Jinaikang) were used in this study. Tetrandrine tablets were obtained from Zhejiang Jinhua CONBA Bio-pharm Co., Ltd. in China. (Approval number: H33022075). According to the drug instructions, most tetrandrine exists in its original form after metabolism in vivo, and a small part is metabolized into tetrandrine-n-oxide isomer and n-2-demethyltetrandrine. All the inpatients in the case group took tetrandrine orally at a dose of 60 mg three times a day, and all those patients were continuously administered tetrandrine for at least one week in hospital and continued for one week after discharge.

### 2.3. Definition

Patients who met discharge criteria and clinical classification followed the Protocol for Diagnosing and Treating Novel Coronavirus Pneumonia (Study Version 7) published by the National Health Commission and the National Administration of Traditional Chinese Medicine. Discharge criteria: (1) Body temperature is back to normal for more than 3 days; (2) respiratory symptoms improve obviously; (3) pulmonary imaging shows obvious absorption of inflammation; (4) nuclei acid tests negative twice consecutively on respiratory tract samples such as sputum and nasopharyngeal swabs (sampling interval being at least 24 h). Clinical classification: Mild cases: The clinical symptoms were mild, and there was no sign of pneumonia on imaging. Moderate cases: Showing fever and respiratory symptoms with radiological findings of pneumonia. Severe cases: Adult cases meeting any of the following criteria: (1) Respiratory distress (≥30 breaths/min); (2) oxygen saturation ≤ 93% at rest; (3) arterial partial pressure of oxygen (PaO_2_)/fraction of inspired oxygen (FiO_2_) ≤ 300 mmHg; and (4) cases with chest imaging that shows obvious lesion progression within 24–48 h >50% shall be managed as severe cases. Child cases meeting any of the following criteria: (1) Tachypnea independent of fever and crying; (2) oxygen saturation ≤ 92% on finger pulse oximeter taken at rest; (3) labored breathing (moaning, nasal fluttering, and infrasternal, supraclavicular, and intercostal retraction), cyanosis, and intermittent apnea; (4) lethargy and convulsion; and (5) difficulty feeding and signs of dehydration. The day 7 improvement rate is calculated via dividing the cumulative number of patients in remission on day 7 by the total number of patients with the symptom.

### 2.4. Statistical Analysis

Continuous variables were expressed as medians or mean and compared by independent Student’s *t*-test or Mann-Whitney U-test. Categorical variables were expressed as percentages and tested with chi-square test or Fisher’s exact test. The primary analysis was a chi-square test with treatment as compared with outcome. Fisher exact tests were used to compare the different outcome between the patients with and without antibiotic, tetrandrine, and arbidol with moderate types. The association between the remission time of clinical symptoms and treatment was compared by the day 7 improvement rate and analyzed by chi-square test. R 4.1.1 and SPSS 23.0 were used for the statistical analysis. *p* < 0.05 were considered as statistically significant.

## 3. Results

### 3.1. Study Population

A total of 60 patients were included in the study, and 83.3% (50/60) of these patients improved after treatment, while 16.7% of the patients (10/60) recovered. The most important patient characteristics are shown in Table 1. The mean age in the improvement groups (46.8 years) and recovery groups (36.8 years) were comparable (*p* = 0.082). The proportion of male patients in the improvement group (54%) was higher than that in the recovery group (30%). Farmers represented the highest proportion (53.3%) of COVID-19 patients; no significant association between occupation and clinical results was found (*p* = 0.215). The BMI in the two groups was equivalent (*p* = 0.486).

Patients with mild types showed a significantly higher recovered proportion than the other groups contained moderate and severe types (*p* < 0.001). The duration of hospitalization was comparable between improvement groups and recovery groups (*p* = 0.758). There were no significant differences between two groups underlying chronic diseases history (*p* > 0.05) and pulmonary disease history (*p* > 0.05). For treatment, there were significant differences between two groups underlying use of antibiotics (*p* < 0.001), tetrandrine (*p* = 0.010), and arbidol (*p* = 0.029). However, there were no significant differences between two clinical outcome groups among the other treatments (*p* > 0.05).

### 3.2. Comparison of the Clinical Outcomes in Total Patients (n = 60) Treated with and without Antibiotic, Tetrandrine, and Arbidol

Of all the 60 COVID-19 patients, antibiotics were used in 98% of the cases in the improvement group and only 50% in the recovery group; the treatment of antibiotic was associated with clinical outcome (*p* < 0.001). No patients were treated with arbidol in the recovery group, and 42% of patients of the improvement group were treated with it. The utilization rate of arbidol was higher in the improvement group than that in the recovery group (*p* = 0.029). Half of patients had treatment with tetrandrine, and the proportion of patients receiving tetrandrine in the recovery groups during hospitalization was higher (100%) than that in improvement groups (40%) (*p* = 0.010) (Table 2).

### 3.3. Comparison of the Clinical Outcomes in Patients with Moderate Type (n = 51) Treated with and without Antibiotic, Tetrandrine, and Arbidol

Patients with mild and severe types only make up a small proportion of the total SARS-CoV-2 infections. The proportion of the elderly in patients with severe type is higher and can lead to more complications, so the same treatment may not be effective in patients with severe type. Mild-type patients have low fever and mild fatigue and usually do not develop pneumonia. Depending on whether patients of these two types are more special and more extreme than moderate patients and whether they can distort the results, patients with the moderate type were also included separately in the analysis.

Although the treatment with and without antibiotic and arbidol were significantly associated with clinical outcome in the total 60 patients, no significant association was shown in the patients with moderate types (*p* > 0.05). When comparing clinical outcome of patients with moderate types with or without tetrandrine, unlike the other two treatments, a significant difference still existed (*p* = 0.007) (Table 3).

### 3.4. Symptom Improvements with and without Antibiotic, Tetrandrine, and Arbidol

Improvement in clinical symptoms of the total 60 COVID-19 patients is presented in Table 4 and Figure 3, and that of patients with moderate type is presented in Table 5 and Figure 4. There was no significant difference in the rate of improvement on day 7 of 4 clinical symptoms (fever, cough, fatigue, and gastrointestinal symptoms) in patients with and without antibiotic and arbidol (*p* > 0.05). The rate of improvement in coughing on day 7 in tetrandrine-treated patients was 100%, which was significantly higher than in patients without tetrandrine treatment (70%) (*p* = 0.003). The rate of improvement in fatigue on day 7 was also 100% in tetrandrine-treated patients and was higher than in patients without tetrandrine treatment (60%) (*p* = 0.003). When the analysis was limited to patients with the moderate type, tetrandrine also significantly improved the rate of improvement in fever and cough on day 7, both of which are 100%, while the rate without tetrandrine was only 70.4% and 59.3%, respectively. A significant association was not shown between the improvement rate on day 7 and the treatment with antibiotics and arbidol (*p* > 0.05).

## 4. Discussion

The ongoing COVID-19 pandemic has caused significant public health damage, leading to constant social panic and enormous economic loss. Especially in some underdeveloped countries and regions, the COVID-19 infection has increased the ongoing burden of child undernutrition [25]. Although many treatment regimens have been tirelessly researched, no specific antiviral therapy has been approved to date. In this study, we found that tetrandrine, previously used as a drug for silicosis, can reduce the time to remission from cough and fatigue and improve the clinical outcome of COVID-19 patients, and these beneficial effects persist in all patients with COVID-19 and patients with the moderate type. Although there were significant differences in clinical outcomes between patients treated with and without antibiotics as well as with arbidol, these differences disappeared when the comparison was restricted to patients with moderate types. Moreover, there was no significant difference in the rate of improvement in clinical symptoms of COVID-19 on day 7 between patients treated with and without antibiotics and arbidol in all patients and patients with the moderate type.

Arbidol, a broad-spectrum antiviral agent also known as umifenovir, is newly added in the Diagnosis and Treatment Protocol for Novel Coronavirus Pneumonia on COVID-19 of Chinese government. However, the effectiveness of arbidol remains divisive. In a retrospective study of 81 COVID-19 patients, an improved outcome or faster clearance of SARS-CoV-2 was not found in the umifenovir group in patients without an intensive care unit [26]. A randomized controlled trial showed that arbidol significantly contributed to clinical and laboratory improvements compared to KALETRA (lopinavir/ritonavir) [27]. In this study, the results were divided into different groups. Significant differences were seen in the total cases, but not in patients with the moderate type. However, the above conclusion needs to be examined because of the limitations of the sample size of the study.

According to some articles, the overuse of antibiotics should be paid more attention owing to the fact that the proportion of COVID-19 patients with a bacterial co-infection is low, and antibiotic therapy is high [28]. In this study, the proportion of recovery was lower in the group treated with antibiotics (*p* < 0.001), and this may be caused by the difference of initial disease severity. In addition, the treatment of antibiotics did not promote the outcome despite 49 (96.1%) patients who came from moderate type receiving antibiotics. The lack of difference may indirectly support the hypothesis regarding antibiotic abuse. In summary, only tetrandrine improved the outcome in two different ways of clustering.

The putative mechanism of action of tetrandrine, which underlies its possible use against COVID-19, can include two aspects: Almost all coronaviruses penetrate host cells by endocytosis, and the study by Ou et al. showed that the entry of SARS-CoV-2 into host cells is primarily mediated by endocytosis [22]. Their study also showed that TPC2 is essential for SARS-CoV-2 to enter, although the specific role of TPC2 in the escape of the virus into the cytoplasm is not entirely clear. Hence, tetrandrine can inhibit COVID-19 replication via blocking TPC2 in host cells. Histopathology analyses showed fibrin deposits in lungs of patients with severe COVID-19, and mild myocardial hypertrophy changes and focal fibrosis are tissue changes seen in the post-mortem heart biopsies of COVID-19 patients [29]. At the middle stage of the disease, alveolar epithelial degeneration, and necrosis as well as varying amounts of cellulose in the alveolar cavity were observed under microscope. The end stage of the disease is marked by fibrotic formation, and the lesions are irreversible [30,31,32]. The mechanism may be related to SARS-CoV2 binding to angiotensin converting enzyme 2 (ACE2) receptors on type II pneumocytes and activating accumulation of fibrin deposits in pulmonary microcapillary venous vessels [33]. By reducing collagen synthesis, tetrandrine can inhibit collagen hyperplasia in lung mesenchymal tissue as well as pulmonary fibrosis, thereby improving the outcome of COVID-19 patients.

In this study, the *p*-value for the rate of fever improvement on day 7 with and without tetrandrine in all COVID-19 patients and moderate-type patients is both marginal and close to the examination level (*p* = 0.056, *p* = 0.053). Therefore, we include fever as an improved clinical symptom to be discussed together. Fever and fatigue are common initial symptoms in people with COVID-19 and can be caused by an inflammatory reaction. The inflammatory storm induced by SARS-CoV-2 causes immune damage and leads to an uncontrolled inflammatory reaction that leads to the production of proinflammatory cytokines [34,35]. Tetrandrine has a broad-spectrum anti-inflammatory effect with complex anti-inflammatory mechanism and includes almost all links of the inflammatory response. Liu et al. found that tetrandrine significantly reduced tumor necrosis factor TNF-α, interleukin IL-1β, and IL-6 in the calvaria [36]. Treatment with tetrandrine decreased the production of TNFα and IL1β and led to downregulation of phosphorylated NF-κB p65 [37]. Its ability to reduce proinflammatory cytokines may explain the reason of the shortening of remission time in fever and fatigue.

Tetrandrine as a potential drug for COVID-19 in previous study is mostly involved pharmacological hypothesis and review, but observational studies in the population are rarely seen. In this study, 60 COVID-19 patients were treated in different therapeutic approaches, and it was found that there were significant differences in outcome between the patients with and without tetrandrine according to the hospital medical records, *p* = 0.010. This result was also available in patients with moderate types, *p* = 0.007. So, tetrandrine as a potential therapeutic for COVID-19 may be clinically effective. There are some limitations to this study, however. The number of samples included in this study is not very large, and the number of patients with mild and severe types is even fewer. Therefore, COVID-19 patients with lung disease are not excluded from this study. Furthermore, time lengths for administration were not enough, with most patients taking tetrandrine for more than 14 days. Because of the individual differences, each patient’s symptoms are slightly different, and the actual treatment regimens except for tetrandrine had some differences that may lead to minor errors in statistical analysis.

Traditional Chinese medicine (TCM) in the treatment of COVID-19 has been a focus of research, with Chinese and South Korean guidelines recommending Chinese herbal medicine as a treatment option for COVID-19 patients. There were three decoctions of TCM and three formulated Chinese medicines that were found to be most effective in treating patients with various stages of COVID-19 in China, including Lianhua Qingwen capsules, Jinhua Qinggan granules, Qingfei Paidu, Huashibaidu granules, Xuebijing, and Xuanfeibaidu granules [38]. A retrospective study of 80 COVID-19 patients showed the pneumonia recovery time in the Jinhua Qinggan group was significantly shorter than the control group [39]. A meta-analysis from Zeng et al. showed the disappearance rate of the main clinical symptoms and other clinical secondary symptoms in the group with Chinese medicine Lianhua Qingwen was significantly higher than that of the control group, and the duration of fever was significantly lower than that of the control group [40]. Xuebijing was recommended by China’s National Health Commission to treat severe/critical patients [38], and a retrospective case-control study found there were significant improvements in body temperature and CT imaging results in the observation group as compared with the control group after treatment, particularly in severe patients [41].

Tetrandrine has a well-documented history as an anti-fibrosis drug and for treating silicosis, and this study may be evidence regarding the effects of tetrandrine for the treatment of COVID-19 patients. Still, the safe and most effective doses that are being administered more widely require more evidence of conclusive pharmacokinetic and toxicological studies and clinical trials. The different administration methods of tetrandrine may lead to different therapeutic effects for inhalation by aerosol and taking it orally; this requires further research as well.

## 5. Conclusions

Among COVID-19 patients, treatment with tetrandrine compared with non- tetrandrine treatment resulted in a higher proportion of recovery and shorter remission time, indicating tetrandrine may be a potential candidate therapeutic agent against COVID-19. Further studies with more well-designed RCTs in COVID-19 treatment with tetrandrine are warranted to verify the effect.

## Figures and Tables

**Figure 1 medicina-58-01194-f001:**
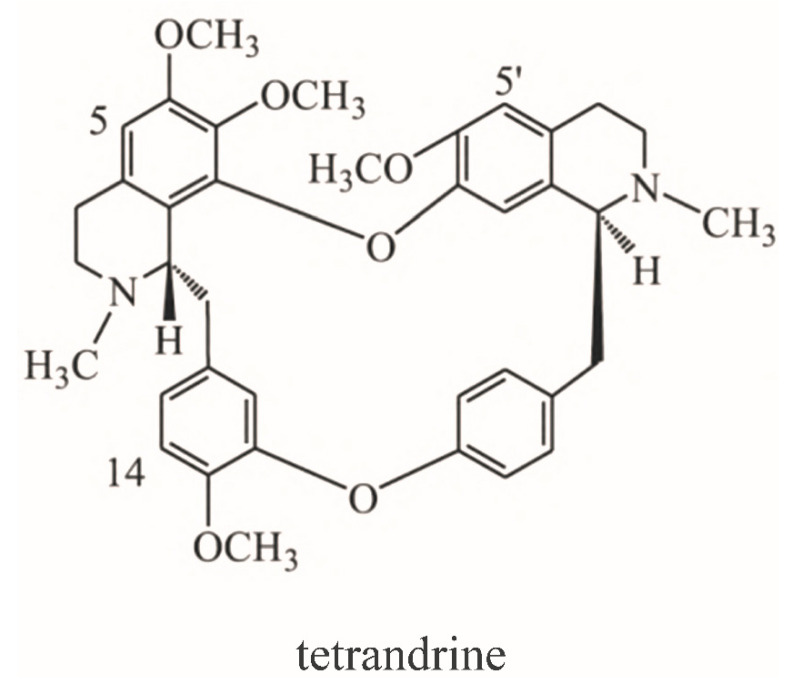
Chemical structure of tetrandrine.

**Figure 2 medicina-58-01194-f002:**
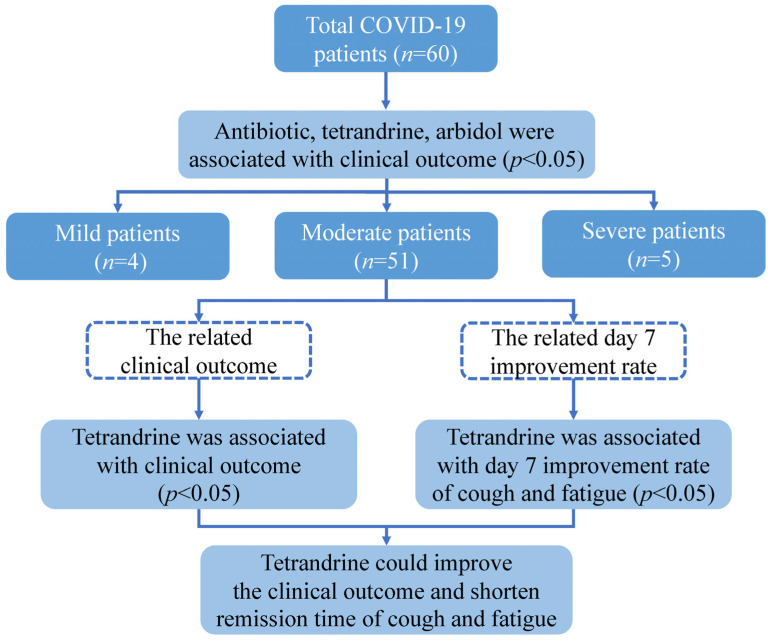
Schematic representation of this study design.

**Figure 3 medicina-58-01194-f003:**
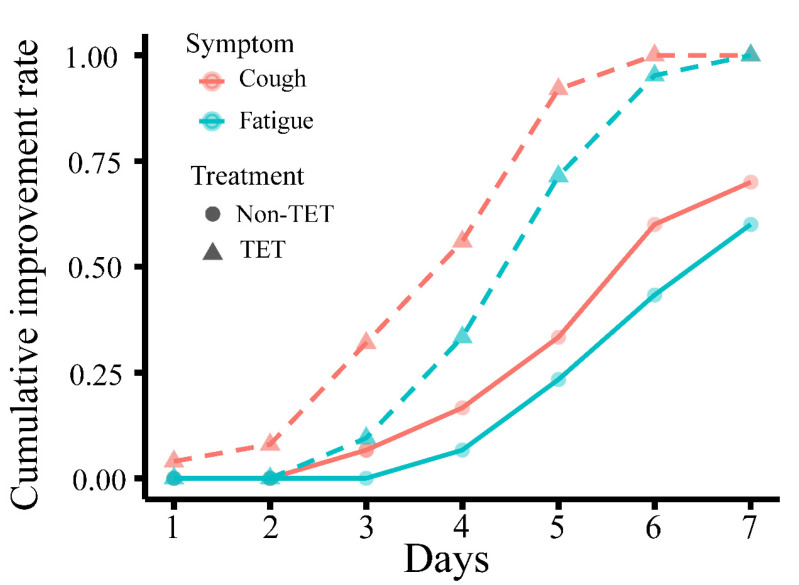
Cumulative day 7 improvement rate of cough and fatigue in COVID-19 patients with and without tetrandrine treatment. The line for cough is orange, and the line for fatigue is green. TET, tetrandrine.

**Figure 4 medicina-58-01194-f004:**
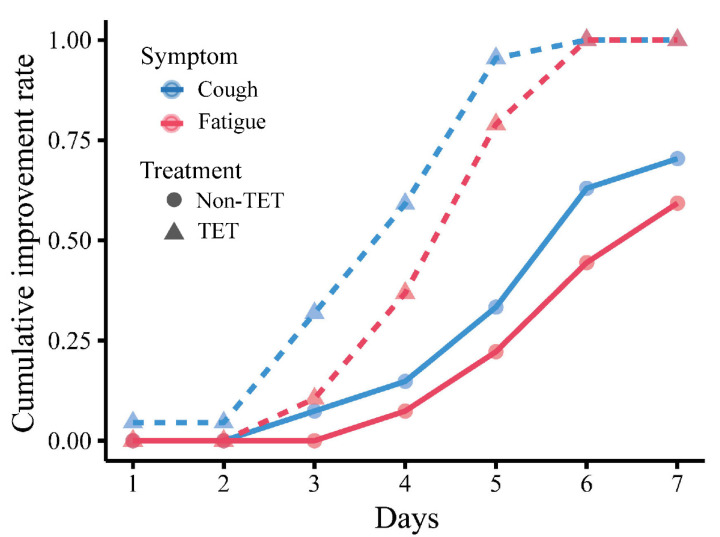
Cumulative day 7 improvement rate of cough and fatigue in moderate COVID-19 patients with and without tetrandrine treatment. The line for cough is red, and the line for fatigue is blue. TET, tetrandrine.

**Table 1 medicina-58-01194-t001:** Demographic and clinical characteristics and treatment comparisons between different outcomes.

Parameters	Improvement (*n* = 50)	Recovery (*n* = 10)	*p*-Value
Age, years	46.8 (5–79)	36.8 (2–69)	0.082
Gender			0.166
Male, *n* (%)	27 (54%)	3 (30%)	
Female, *n* (%)	23 (46%)	7 (70%)	
Occupation			0.215
Worker, *n* (%)	9 (18%)	1 (10%)	
Farmer, *n* (%)	26 (52%)	3 (30%)	
Office clerk, *n* (%)	4 (8%)	2 (20%)	
Student, *n* (%)	2 (4%)	1 (10%)	
Other, *n* (%)	9 (18%)	3 (30%)	
BMI			0.486
<24 kg/m^2^, *n* (%)	26 (52%)	7 (70%)	
>24 kg/m^2^, *n* (%)	24 (48%)	3 (30%)	
Smoke, *n* (%)	3 (6%)	0 (0%)	1
Initial disease severity			<0.001
Mild, *n* (%)	0 (0%)	4 (40%)	
Moderate/severe, *n* (%)	50 (100%)	6 (60%)	
Duration of hospitalization, days	12 (6–25)	12 (9–21)	0.758
Chronic diseases history			
Hypertension, *n* (%)	9 (18%)	1 (10%)	0.877
Diabetes, *n* (%)	2 (4%)	0 (0%)	1
Hepatitis, *n* (%)	3 (6%)	1 (10%)	0.528
Coronary heart disease, *n* (%)	5 (10%)	1 (10%)	1
Pulmonary disease history, *n* (%)			
Tracheitis/chronic Bronchitis/asthma, *n* (%)	6	2	0.865
Bronchiectasis, *n* (%)	2	0	1
Obsolete pulmonary tuberculosis, *n* (%)	4	0	1
Treatment			
Tetrandrine, *n* (%)	20 (40%)	10 (100%)	0.010
Arbidol, *n* (%)	21 (42%)	0 (0%)	0.029
Antibiotic, *n* (%)	49 (98%)	5 (50%)	<0.001
Hormone, *n* (%)	8 (16%)	0 (0%)	0.396
Reduce phlegm, *n* (%)	38 (76%)	8 (80%)	1
Supportive treatment, *n* (%)	2 (4%)	0 (0%)	1
Oxygen therapy, *n* (%)	13 (26%)	2 (20%)	1
Ribavirin, *n* (%)	30 (60%)	9 (90%)	0.146
Vitamin C, *n* (%)	28 (56%)	5 (50%)	0.742
Interferon, *n* (%)	42 (84%)	10 (100%)	0.396
Lianhua Qingwen, *n* (%)	13 (26%)	0 (0%)	0.161

**Table 2 medicina-58-01194-t002:** Effect on outcome of treatment with antibiotic, tetrandrine, and arbidol of total COVID-19 patients (*n* = 60).

Treatment	Improvement	Recovery	Sum	*p*-Value
Antibiotic	49 (98%)	5 (50%)	54 (90%)	0.001
Non-antibiotic	1 (2%)	5 (50%)	6 (10%)	
TET	20 (40%)	10 (100%)	30 (50%)	0.010
Non-TET	30 (60%)	0 (0%)	30 (50%)	
Arbidol	21 (42%)	0 (0%)	21 (35%)	0.029
Non-arbidol	29 (58%)	10 (100%)	39 (65%)	

**Table 3 medicina-58-01194-t003:** Effect on outcome of treatment with antibiotic, tetrandrine, and arbidol of moderate COVID-19 patients (*n* = 51).

Treatment	Improvement	Recovery	Sum	*p*-Value
Antibiotic	44 (97.8%)	5 (83.3%)	49 (96.1%)	0.224
Non-antibiotic	1 (2.2%)	1 (16.7%)	2 (3.9%)	
TET	18 (40.0%)	6 (100%)	24 (47.1%)	0.007
Non-TET	27 (60.0%)	0 (0%)	27 (52.9%)	
Arbidol	13 (28.9%)	0 (0%)	13 (25.5%)	0.318
Non-arbidol	32 (71.1%)	6 (100%)	38 (74.5%)	

**Table 4 medicina-58-01194-t004:** Improvement of clinical symptoms in total COVID-19 patients with and without antibiotic, tetrandrine, and arbidol.

	Day 7 Improvement Rate		*p*-Value	Day 7 Improvement Rate		*p*-Value	Day 7 Improvement Rate		*p*-Value
Symptom	Antibiotic	Non-Antibiotic		TET	Non-TET		Arbidol	Non-Arbidol	
Fever	87.5% (35/40)	100% (1/1)	1	100% (18/18)	78.3% (18/23)	0.056	85.7% (6/7)	88.2% (30/34)	1
Cough	82.7% (43/52)	100% (3/3)	1	100% (25/25)	70.0% (21/30)	0.003	76.9% (10/13)	85.7% (36/42)	0.749
Fatigue	76.0% (38/50)	100% (1/1)	1	100% (21/21)	60.0% (18/30)	0.003	76.9% (10/13)	76.3% (29/38)	1
Gastrointestinal symptoms	86.8% (33/38)	100% (1/1)	1	100% (15/15)	79.2% (19/24)	0.136	77.8% (7/9)	90.0% (27/30)	0.694

**Table 5 medicina-58-01194-t005:** Improvement of clinical symptoms in moderate COVID-19 patients with and without antibiotic, tetrandrine and arbidol.

	Day 7 Improvement Rate		*p*-Value	Day 7 Improvement Rate		*p*-Value	Day 7 Improvement Rate		*p*-Value
Symptom	Antibiotic	Non-Antibiotic		TET	Non-TET		Arbidol	Non-Arbidol	
Fever	87.5% (30/35)	100% (1/1)	1	100% (16/16)	75.0% (15/20)	0.053	85.7% (6/7)	86.2% (25/29)	1
Cough	83.0% (39/47)	100% (2/2)	1	100% (22/22)	70.4% (19/27)	0.006	76.9% (10/13)	86.1% (31/36)	0.741
Fatigue	75.6% (34/45)	100% (1/1)	1	100% (19/19)	59.3% (16/27)	0.005	76.9% (10/13)	75.8% (25/33)	1
Gastrointestinal symptoms	87.9% (29/33)	100% (1/1)	1	100% (13/13)	81.0% (17/21)	0.144	77.8% (7/9)	92.0% (23/25)	0.281

## Data Availability

The data presented in this study are available on request from the corresponding author.

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
