# Peer review of "Tetrandrine Treatment May Improve Clinical Outcome in Patients with COVID-19"

_medicina, 2022, doi:10.3390/medicina58091194_

Round 1

Reviewer 1 Report

Dear Editor,

After reading this manuscript, I agree that the language of the manuscript is good. I also have several questions, as shown below:

1. The authors only wrote a Hubei hospital. This should be clearly presented. Which is the accurate hospital for performing this experiment?

2. The data is not enough, either indexes or time lengths for administration of the medicines. The manuscript generally lacks broadness and depth.

Reviewer 2 Report

1| L. 60-61. Statisctical data should be updated, e.g. for August, 1 , 2022.

2| Provide briefly data concerning origin and chemical structure of tetrandrine

3| Doses, mode and duration of administration of tetrandrine or their ranges should be mentioned. Also authors should clarify whether tetrandrine comprises the total extract of the herb or an individual compound

4| Captures for the fig. 2 and 3 should contain definition for the abbreviation TET

Round 2

Reviewer 1 Report

The authors made adequate revisions and I recommend its publication in Medicine.

Reviewer 2 Report

As a reviewer of the manuscript, I am satisfied by all proposed corrections made by the authors in the updated version.